# Feasibility and usability of a very low-cost bubble continuous positive airway pressure device including oxygen blenders in a Ugandan level two newborn unit

Anna B. Hedstrom[1]*, James Nyonyintono[2], Eugene A. Saxon[3], Heidi Nakamura[4], Hilda Namakula[4], Beatrice Niyonshaba[4], Josephine Nakakande[2], Noelle Simpson[4], Madeline Vaughan[4], Alec Wollen[5], Paul Mubiri[6], Peter Waiswa[6], Patricia S. Coffey[5], Maneesh Batra[1]

1 Department of Pediatrics and Global Health, University of Washington, Seattle, Washington, United States of America, 2 Kiwoko Hospital, Kiwoko, Uganda, 3 EPFL EssentialTech Centre, the Swiss Federal Institute of Technology, Lausanne, Switzerland, 4 Adara Development, Sydney, Australia, 5 Medical Devices and Health Technologies, PATH, Seattle, Washington, United States of America, 6 Department of Health Policy, Planning and Management, Makerere University, Kampala, Uganda

* hedstrom@uw.edu

## Abstract

### Background

Preterm birth and resulting respiratory failure is a leading cause of newborn death- the majority of which occur in resource-constrained settings and could be prevented with bubble continuous positive airway pressure (bCPAP). Commercialized devices are expensive, however, and sites commonly use improvised devices utilizing 100% oxygen which can cause blindness. To address this, PATH and a multidisciplinary team developed a very low-cost bCPAP device including fixed-ratio oxygen blenders.

### Objective

We assessed feasibility of use of the device on neonatal patients as well as the usability and acceptability of the device by healthcare workers. This study did not evaluate device effectiveness.

### Methods

The study took place in a Ugandan level two unit. Neonates with respiratory failure were treated with the bCPAP device. Prospective data were collected through observation as well as likert-style scales and interviews with healthcare workers. Data were analyzed using frequencies, means and standard deviation and interviews via a descriptive coding method. Retrospectively registered via ClinicalTrials.gov number NCT05462509.

### Results

Fourteen neonates were treated with the bCPAP device in October—December 2021. Patients were born onsite (57%), with median weight of 1.3 kg (IQR 1–1.8). Median

**Data Availability Statement:** The dataset is included in the Supporting Information files.

**Funding:** Funding provided by Saving Lives at Birth (https://savinglivesatbirth.net/) Awd #1802-32086 to PATH. Initials of the authors who were funded under this award: AH, JN, HNakamura, HNamakula, BN, JN, AW, PM, PSC. This project is made possible through the generous support of the Saving Lives at Birth partners: the United States Agency for International Development (USAID), the Norwegian Agency for Development Cooperation (Norad), the Bill & Melinda Gates Foundation, Grand Challenges Canada, the U.K. Department for International Development (DFID), and the Korea International Cooperation Agency (KOICA). This manuscript was prepared by the University of Washington, Adara Development, Kiwoko Hospital, PATH and Makerere University and does not necessarily reflect the views of the Saving Lives at Birth partners. The funders had no role in study design, data collection and analysis, decision to publish, or preparation of the manuscript.

**Competing interests:** The authors have declared that no competing interests exist.

treatment length was 2.5 days (IQR 2–6). bCPAP was stopped due to: improvement (83%) and death (17%). All patients experienced episodes of saturations >95%. Median time for device set up: 15 minutes (IQR 12–18) and changing the blender: 15 seconds (IQR 12–27). After initial device use, 9 out of 9 nurses report the set-up as well as blender use was "easy" and their overall satisfaction with the device was 8.5/10 (IQR 6.5–9.5). Interview themes included the appreciation for the ability to administer less than 100% oxygen, desire to continue use of the device, and a desire for additional blenders.

## Conclusions

In facilities otherwise using 100% oxygen, use of the bCPAP device including oxygen blenders is feasible and acceptable to healthcare workers.

## Trial registration

ClinicalTrials.gov, Identifier NCT05462509.

## Background

Over two million newborns die each year globally, the majority in low resource settings [1]. The final common pathway in the most frequent causes of neonatal death- preterm birth, infection, and birth asphyxia- is respiratory failure [1–3]. Among premature infants, an estimated 45% of deaths are due to respiratory distress syndrome from surfactant deficiency which makes it difficult to maintain adequate lung expansion [2].

Continuous positive airway pressure (CPAP) provides constant distending pressure to the lungs and decreases the work of breathing for an infant. It is standard of care for respiratory distress syndrome and decreases mortality in premature infants [4–6]. One type of CPAP, bubble CPAP (bCPAP), is created by submerging an expiratory tube in water. bCPAP is considered superior in treating neonatal respiratory failure compared with flow driven CPAP because it results in lower failure rates and need for mechanical ventilation [7]. This is particularly important in settings without the advanced support of ventilators. Commercialized bCPAP devices are available for several thousand US dollars yet are rarely found in resource constrained settings due to this prohibitive cost [8–10].

Commercialized devices considered to be "low cost" do exist, however, many of these devices remain too expensive for sufficient supply in resource-constrained settings (RCS), with costs approaching or exceeding US$1,000 [8, 11–14]. To address this care gap, the World Health Organization (WHO) provides guidance on creation of locally made, or improvised bCPAP devices assembled at the point of care from components available in the hospital [15]. These improvised devices, however, have not been bench tested for reliability and their performance is variable, particularly in terms of pressure delivered to the lung due to the variation in the length and bore of tubing used, small cannula sizes, and incorrect assembly of parts [16, 17].

Additionally, most RCS providing bCPAP therapy do not have a mechanism to provide blended gas (air and oxygen) to the newborn and instead use 100% oxygen [9, 10, 18]. WHO guidelines strongly advise against the use of 100% oxygen with newborns, as it poses a risk for blindness, chronic lung disease and brain injury [15, 19, 20]. High resource settings use a commercialized oxygen blender to avoid these risks. However, these blenders are expensive at about US$1,000 each, and require pressurized sources of both oxygen and air, the latter of which is rarely available in RCS. As preterm infants in RCS settings increasingly receive treatment with bCPAP, evidence suggests that blindness due to exposure to oxygen is increasing [21].

There is a critical need for a safe, rigorously tested, very low-cost bCPAP device including oxygen blenders for resource-constrained sites [3, 22, 23]. The effective use of such devices in referral and district hospitals could prevent 178,000 neonatal deaths in Africa each year [24]. To address this need, PATH worked with a multidisciplinary team and developed a very low-cost bCPAP device including two fixed-ratio oxygen blenders that do not require a source of pressurized air [25–27]. The effective implementation and use of such a device, however, requires investigation into the practicality of its use by healthcare workers [28].

We conducted an early feasibility study of this bCPAP device including oxygen blenders with neonates in respiratory failure. The study took place in a rural level two newborn care unit in Uganda with experience using bCPAP and blended oxygen. We hypothesized that the device would be acceptable to, and easy to use by healthcare workers and allow patients to receive less than 100% oxygen when no other blending mechanism was available. Our primary objectives were to assess the operational feasibility of the device as well as its usability and acceptability by healthcare workers. Our secondary objective was to report clinical characteristics, demographics and outcomes of patients treated with the device and blenders.

## Methods

### Study design

This was an early feasibility study using mixed methods to collect both qualitative data from healthcare worker experience as well as quantitative data from device use.

### Period of study

Data were collected from October 2021 through January 2022. Enrollment was paused 10 November to 21 November as study staff were unavailable.

### Study setting

The Kiwoko newborn care unit is a level two newborn unit in a rural, private, not-for-profit hospital. The unit admits neonates below 44 weeks corrected gestational age and acts as a referral center for three districts in central Uganda. There were 1,276 admissions in 2021. Thirty-five nurses are on staff with five to seven on duty each shift, as well as one medical officer and one pediatrician. Electricity is constantly available. The unit provides thermoregulation via overhead warmers, incubators and kangaroo care, infection control and treatment, intravenous hydration, nasogastric and cup feeding, phototherapy, blood transfusion, health education, neurodevelopmental supportive care, basic laboratory services and respiratory therapies described below. The unit does not have access to total parenteral nutrition.

### Pre-existing respiratory therapies

Pre-existing respiratory therapies included aminophylline, oxygen from concentrators via nasal cannula and improvised bCPAP (described below), and intermittent or continuous pulse oximetry. Neither mechanical ventilation nor surfactant is available. If needed, transport is offered free of charge for referral to a unit where mechanical ventilation and surfactant are available, however hospitalization costs at referral units are prohibitively expensive for much of the population served by Kiwoko Hospital.

### Pre-existing CPAP

Improvised bCPAP delivery via a device, assembled on-site using donated nasal cannulas, has been the standard of care for respiratory failure in the unit since 2012 [29]. There is no limit on

how many infants can be treated simultaneously with bCPAP in the unit and no restrictions in diagnosis nor birthweight/gestational age for use of bCPAP. It is used as the highest level of respiratory support for any patient in need. Historically, fifteen percent of patients admitted to the Kiwoko unit are treated with bCPAP therapy with a median treatment length of 3 days (data from unit database). The baseline mortality for patients treated with bCPAP since 2012 was 34%.

### Pre-existing oxygen blending

Prior to this study, patients had intermittent access to blended oxygen via Y-in of flow from a very limited availability of air compressors. This method allows for delivery of blended oxygen, titrated to any specified amount between 21 and 100%. These compressors, however, fail frequently when used continuously over several days. Due to limited availability of working air compressors, Kiwoko patients may receive bCPAP therapy with up to 100% oxygen, which can put them at risk of oxygen related toxicity.

### bCPAP therapy during study

Prior to study initiation, nurses received two days of intensive training, which included refresher training on bCPAP, as well as use of the new device. During the study, bCPAP therapy and oxygen blending were used according to unit guidelines and physician/nurse discretion based on assessment of the degree of respiratory failure using the Silverman Andersen score (S1 File) [30]. The local practice guideline suggests a Silverman Andersen score of 5 and above as criteria for bCPAP initiation. Starting pressure is 5cm $H_2O$ with maximum suggested pressures of 8cm $H_2O$. Flow rates were adjusted to achieve gentle, continuous bubbling during the respiratory cycle without causing vigorous bubbling. In the absence of apnea, the bCPAP was weaned and then stopped as the respiratory score improved. Patients were monitored with continuous pulse oximetry. The RAM cannula (Neotech) were sized to fill approximately 80% of the nares. Skin safe adhesive was provided with the device to secure the RAM nasal cannula to the face. Nasal foam pads (Neotech NeoSeal) adhering to the nasal prongs were used to minimize pressure loss and tissue damage around the nares. Nasal normal saline drops were used as necessary to prevent mucosal drying.

### Study population

Patients- All newborns admitted to the unit with respiratory failure requiring treatment with bCPAP therapy whose caregiver provided consent were enrolled. Those not enrolled were treated with the existing improvised bCPAP and received oxygen blending when the air compressor was available.

Healthcare workers- Nurses, midwives and doctors in the unit who cared for patients on the novel bCPAP device during the period of study and consented to participation were enrolled in the study.

### Sample sizes

Patients- The sample size for patients was purposive to allow for reasonable frequency of use of the device by healthcare workers in an early feasibility study.

Healthcare workers- The sample size for observed health care workers comprised the workers who cared for the patients receiving bCPAP therapy during the period of study. The target interview sample size was 15 due to the study's limited scope and high level of specificity in the nature of questioning regarding the acceptability and feasibility of the device in a clinical setting [31]. Interview participants were recruited through purposive sampling to include those with a range of experience using the bCPAP device at Kiwoko hospital.

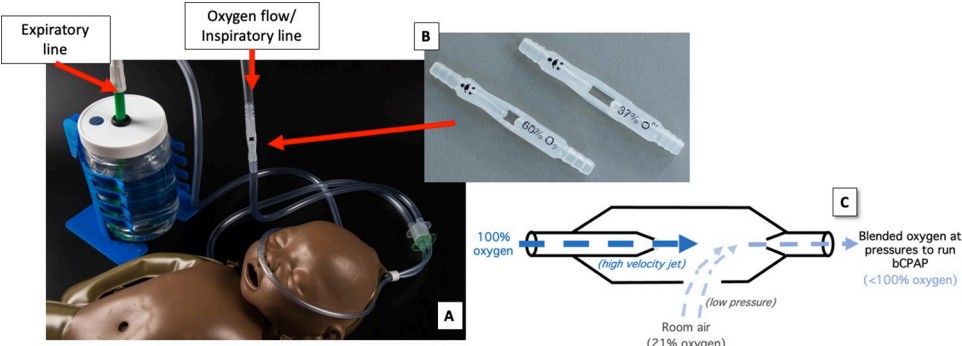

**Fig 1. Very low-cost bCPAP device for neonates. (A)** The system's expiratory tubing is submerged under water to control pressure delivery measured in cm $H_2O$. Patient interface is a RAM cannula (Neotech). The device does not require electricity. **Oxygen blenders (B)** of 37% or 60% can be placed interchangeably within the inspiratory line of the system, in response to the patient's oxygen requirement. **Mechanism of oxygen blender (C)** which functions like a Venturi mask. Room air is entrained into the stream of oxygen via Bernoulli's principle and delivers a precise blend of oxygen and air to the patient. Three to four LPM of oxygen flow from the source is adequate to maintain pressure to the patient due to entrained air which contributes to the total flow.

## Materials

### bCPAP device including oxygen blenders

PATH developed a novel low-cost bCPAP device in 2015 consisting of 5 mm wide tubing, two interchangeable blenders, fittings, RAM cannula (Neotech), and transparent, adjustable pressure-generating bubbler bottle (Fig 1). Device bench testing is described in Coffey 2022 and includes respiratory circuit verification, blender verification, conditioned humidity testing, and sound measurement [27].

The two, small, fixed-ratio oxygen blenders (37% and 60% oxygen ratios respectively) were manufactured using medical grade polyethylene and injection molded using standard equipment. No mold release was used to avoid introduction of contaminants. These devices were designed to provide a stable oxygen blend independent of flow rate and pressure. The chosen blender is placed in-line between the pressurized oxygen source and the patient and functions like a Venturi mask. As oxygen passes through the blender, room air is entrained at a fixed ratio to create a blended gas of either 37% oxygen or 60% oxygen depending on the blender chosen [32]. This mechanism is based on the Bernoulli principle. In this system, the final percent oxygen delivered to the newborn depends on the concentration of oxygen supplied to the blender (e.g. using lower purity oxygen concentrators will result in lower blends delivered to the patient) but remains within ±5% of the blenders, nominal blend ratio for the full range of clinically relevant pressure (4 to 8 cm$H_2O$) and despite change in flows (2 to 7 LPM) for newborns [27]. A minimum of 8.5 psi oxygen source pressure is recommended for blender operation, which is achievable by many medium- or large-sized oxygen concentrators. No electricity is required for the device to operate when used with an oxygen cylinder. The device provides passive humidity provided by entrained ambient air. During the study additional humidity was provided by a 'bubbler' humidifier on the oxygen concentrator. Use of the bubble humidifier limits the oxygen source pressure depending on safety valve setting (commonly 3–6 psi), which in turn reduces the maximum blended flow output capability of the blender. This device is intended to provide respiratory support to a neonate in respiratory distress where higher level means of respiratory support are not available. (See S2 File for instructions for use). The devices were manufactured and performance tested by Sinapi biomedical, an ISO 13485:2016-certified medical device manufacturer based in Stellenbosch, South Africa (http://sinapibiomedical.com/).

---

**bCPAP Device Characteristics**

Tubing diameter- 5mm diameter

Nasal interface- Short binasal cannula (Ram cannula, Neotech)

Humidity- Passive. The blender entrains room air with ambient humidity. In addition, the oxygen source (concentrator or cylinder) may provide additional humidity via a bubbler bottle.

Single Use

Oxygen source requirements: 10LPM at 20psi for maximum blender performance, generally requires below 5LPM at 8.5 PSI for low birthweight patients

Blending range: Fixed-ratio oxygen blenders of 37% and 60%

Electricity: none required

Cost of parts: 19 USD

---

### Oxygen blending

When available, an air compressor remained the standard of care and first choice for blending during the study, allowing full titration of oxygen delivery between 21% and 100%. When an air compressor was not available, enrolled patients received blending using the fixed ratio blenders. Oxygen was titrated based on continuous pulse oximetry. Goal saturation ranges were 90–95% for patients 32 weeks and/or greater and above 1250 grams and 88–95% for those less than 32 weeks and below 1250 grams [15]. For patients with saturations above their goal range, despite use of the lowest blender of 37%, a concentrator that produced a lower percentage of oxygen was used when available. Oxygen concentrators were tested using an oxygen analyzer to assess the percentage of oxygen they provided.

### Participant consent

For patients, initial verbal consent was obtained from a parent or legal guardian at the time of initiation of study device and followed within 24 hours by written, informed consent in English or Luganda. Verbal consent with a witness instead of written consent was used when indicated. For health care workers, informed consent was obtained prior to observation and before interviews.

### Patient data collection

Research assistants were present 24 hours a day in the unit with the sole purpose of data collection of study patients and observation of healthcare workers. They observed the care of bCPAP patients and recorded data on patient clinical characteristics, vital sign/bCPAP/oxygen parameters and the use characteristics of bCPAP. They did not provide patient care but could relay any safety concerns to the bedside healthcare workers.

### Health worker data collection

Healthcare workers completed a likert-type usability scale immediately after device set-up. A subset participated in semi-structured interviews after use of the device related to device usability and acceptability. The interview guide was designed by a pediatrician and public

---

health professional with experience in development and research of health technologies in low resource settings. The goal of the interview guide was to complement the patient data and paint a broader context and story behind the feasibility and acceptability of the bCPAP device. The interview guides focused on the following areas of bCPAP use and oxygen blending: advantages/ disadvantages to the device, things most liked/disliked, ease-of-use, adequacy of blends and what the HCW would change about the device. See S3 File for full interview guide.

The interviews were conducted within a 30–45 minute time frame by a Kiwoko research team member within the hospital environment, and outside of the newborn unit. The research team member knew the participants, however, she worked within the hospital community in a non-clinical role. Interviews were completed in English and transcribed.

## Adverse events

Patients were monitored for the following adverse events: nasal irritation or breakdown, abdominal distention or feeding intolerance, hypoxia (saturations below goal for >5 minutes despite intervention), hyperoxia (saturations above goal for >5 minutes despite intervention), pneumothorax and death while on bCPAP. Events were recorded by research assistants and reported to study supervisors for verification, then reviewed by the medical monitoring committee. This committee included two Ugandan physicians with experience caring for neonates with bCPAP and one biomedical engineer with experience in the administration of oxygen systems. Criteria for stopping the study included an increase in mortality of patients on bCPAP by two standard deviations from baseline at Kiwoko or a pneumothorax rate of 2 out of 10 patients treated.

## Study costs

As standard of care, all neonates who require bCPAP therapy can access it regardless of the parent's financial means. bCPAP therapy is considered a part of basic treatment and is not an extra charge.

## Data analysis

Quantitative data were collected using case report forms and entered in REDCap software and transferred to Stata V.15 for analysis [33, 34]. Data validation rules including consistency checks were included to ensure high quality data. Descriptive analysis using means, standard deviation, median and interquartile range were used to summarize continuous variables and frequencies and percentage used for categorical variables.

Interview data analysis was carried out by two individuals from the research team- the interviewer positionality is described in S4 File. All interviews were transcribed and analyzed using a descriptive coding method. Per Saldana, descriptive coding is "appropriate for virtually all qualitative studies" and "summarizes words or short phrases" to establish the theme of the qualitative data source [35]. After coding was completed, a content analysis framework was used to organize the data and is described in S4 File [36].

Data analysis and coding software was not utilized throughout this process but rather the data was manually coded and reviewed. Recognizing the importance of researcher positionality in qualitative research, the research team was reflective in each iteration of the analysis to minimize the influence of biases in the interpretation of the data.

## Ethics approval

Ethics approval was obtained from the Makerere University School of Public Health Higher Degrees, Research and Ethics Committee (#790) on 5 May 2020, Ugandan National Council

for Science and Technology (HS650ES) on 18 July 2021 as well as the PATH Research and Ethics Committee (1519912–6) on 26 July 2021. The Ugandan Ministry of Health was kept up to date on device development and study planning via presentation to the National Newborn Steering Committee. The study was retrospectively registered via ClinicalTrials.gov, Identifier NCT05462509 and the protocol is included in S6 File.

## Results

### Patient demographics and device use

Fifty-seven patients were treated with bCPAP at Kiwoko hospital during the study period. 14 of these patients were enrolled in the study and treated with the study bCPAP device (Fig 2). Patients not enrolled and therefore treated with the existing bCPAP included: 27 who did not have consent available within 24 hours of bCPAP initiation, 5 for whom families/guardians declined consent and 11 initiated on bCPAP during an enrollment break from 10 November to 21 November. No patients were transferred for respiratory care elsewhere during the period of study nor discharged against medical advice.

Patient characteristics are described in Table 1. Of the 14 patients enrolled, 4 were born via cesarean section. Median apgars scores were 7 (IQR 7–8) and 8 (IQR 8–9) at 1 and 5 minutes respectively. Prior to starting CPAP, 11/14 patients were on nasal cannula oxygen and 10/12 patients for whom it was recorded had a saturation >95%. One air compressor was available for blending one patient at a time throughout study and was used preferentially as dictated by study protocol. Twelve of 14 patients (85.7%) were treated with the blenders during their course and the remaining 2/14 used an available air compressor instead.

### Clinical courses

All patients were started on a CPAP pressure of 5 cm $H_2O$ (Fig 3). Median peak pressure was 6cm $H_2O$ (IQR 5–6). Median peak oxygen flow needed with a blender in place was 4 LPM

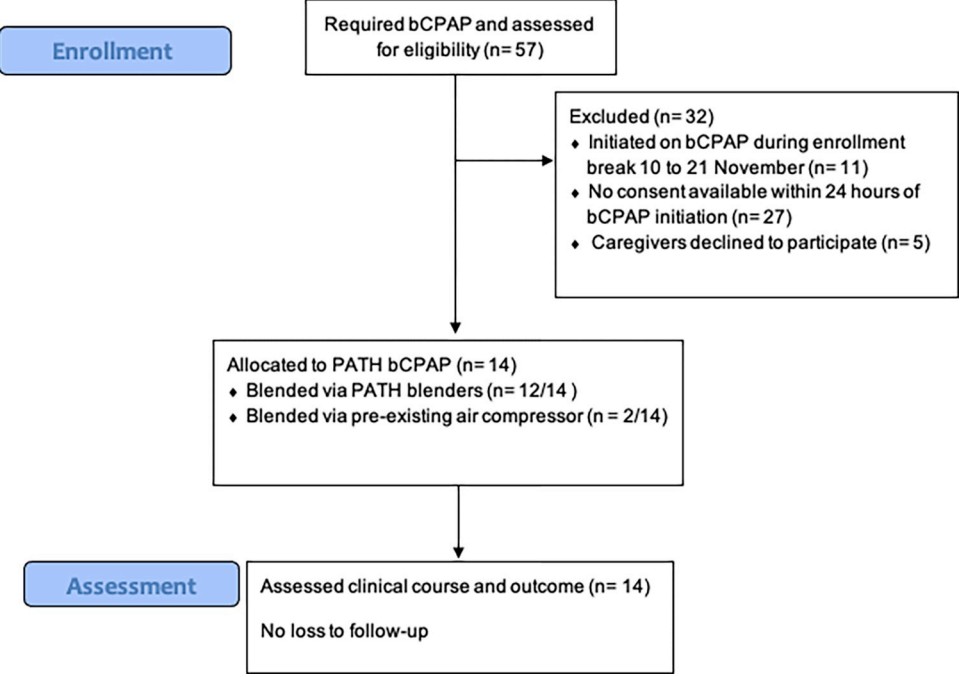

**Fig 2. CONSORT flow diagram for feasibility trial.**

**Table 1. Clinical characteristics and bCPAP courses of participants.**

| | Gest. (weeks) | Admit weight (grams) | Sex | Birth location | Respiratory severity score before bCPAP | Age at bCPAP start (days) | Course length (days) | Diagnoses | Reason for stopping bCPAP | Length of hospital stay (days) | Disposition |
|---|---|---|---|---|---|---|---|---|---|---|---|
| 1 | 25 | 870 | Male | Onsite | 6 | 0 | 12 | Extreme prematurity and respiratory distress | Clinical improvement | 46 | Home |
| 2 | 32 | 1430 | Female | Onsite | 5 | 0 | 6 | Prematurity and respiratory distress | Clinical improvement | 37 | Home |
| 3 | unk | 1900 | Male | On way to hospital | 5 | 0 | 2 | Prematurity, hypoglycemia and severe hypothermia | Clinical improvement | 27 | Home |
| 4 | 27 | 990 | Female | Onsite | unk | 22 | 2 | Extreme prematurity and hypothermia | Clinical improvement | 51 | Home |
| 5 | unk | 2400 | Male | Outside facility | 5 | 0 | 3 | Birth asphyxia, transient tachypnea | Clinical improvement | 13 | Home |
| 6 | 34 | 1900 | Male | Onsite | 5 | 0 | 2 | Prematurity, respiratory distress, hypoglycemia and severe hypothermia | Clinical improvement | 10 | Home |
| 7 | 32 | 1190 | Female | Onsite | 3 | 1 | 18 | Prematurity | Clinical improvement | 49 | Home |
| 8 | 32 | 1400 | Female | Onsite | 3 | 1 | 3 | Prematurity and respiratory distress | Clinical improvement | 30 | Home |
| 9 | 25 | 800 | Male | Onsite | 6 | 0 | 6 | Extreme prematurity and respiratory distress | Death | 6 | Died |
| 10 | 31 | 1320 | Male | Outside facility | 8 | 0 | 1 | Prematurity and respiratory distress | Clinical improvement | 39 | Home |
| 11 | 35 | 1870 | Female | Outside facility | unk | 1 | 2 | Prematurity and respiratory distress | Clinical improvement | 16 | Home |
| 12 | 32 | 1300 | Male | Onsite | 4 | 0 | 2 | Prematurity | Clinical improvement | 33 | Home |
| 13 | 29 | 1050 | Female | Outside facility | 5 | 0 | 6 | Prematurity and respiratory distress | Death | 6 | Died |
| 14 | 26 | 850 | Female | Outside facility | 3 | 7 | 3 | Extreme prematurity and respiratory distress | Clinical improvement | 41 | Home |
| **Median** | **31.5** | **1310g** | **50% female** | **57% born onsite** | **5** | **0 days** | **3 days** | **93% prematurity** | **85% improved** | **31.5 days** | **85% Home** |
| (IQR) | (26–32) | (1005–1760) | | | (4–5) | (0–1) | (2–6) | | | (14–40) | |

Gest = gestation. Unk = unknown

(IQR 3.5–5) and without the blender in place 5 LPM (IQR 5–6). Median pressure changes per day were 0.3 (IQR 0–0.9). Overall, the 37% blender was in use 90% of the time for patients requiring treatment with the blenders. Median blender changes per day were 0.3 (IQR 0.08–1). Median treatment length was 3 days (IQR 2–6). bCPAP was stopped due to: clinical improvement (12/14), death (2/14) and for no patients due to complications. The surviving 12 patients were subsequently discharged home after a median hospital stay of 34 days (IQR 23–41).

Adverse events included: episodes of hyperoxia (oxygen saturations >95% lasting greater than 5 minutes) in 14/14 patients and death (including progressive hypoxia) in 2/14 patients. No occurrences of nasal irritation, abdominal distention, nasal breakdown, feeding intolerance nor pneumothorax. No device-related events occurred.

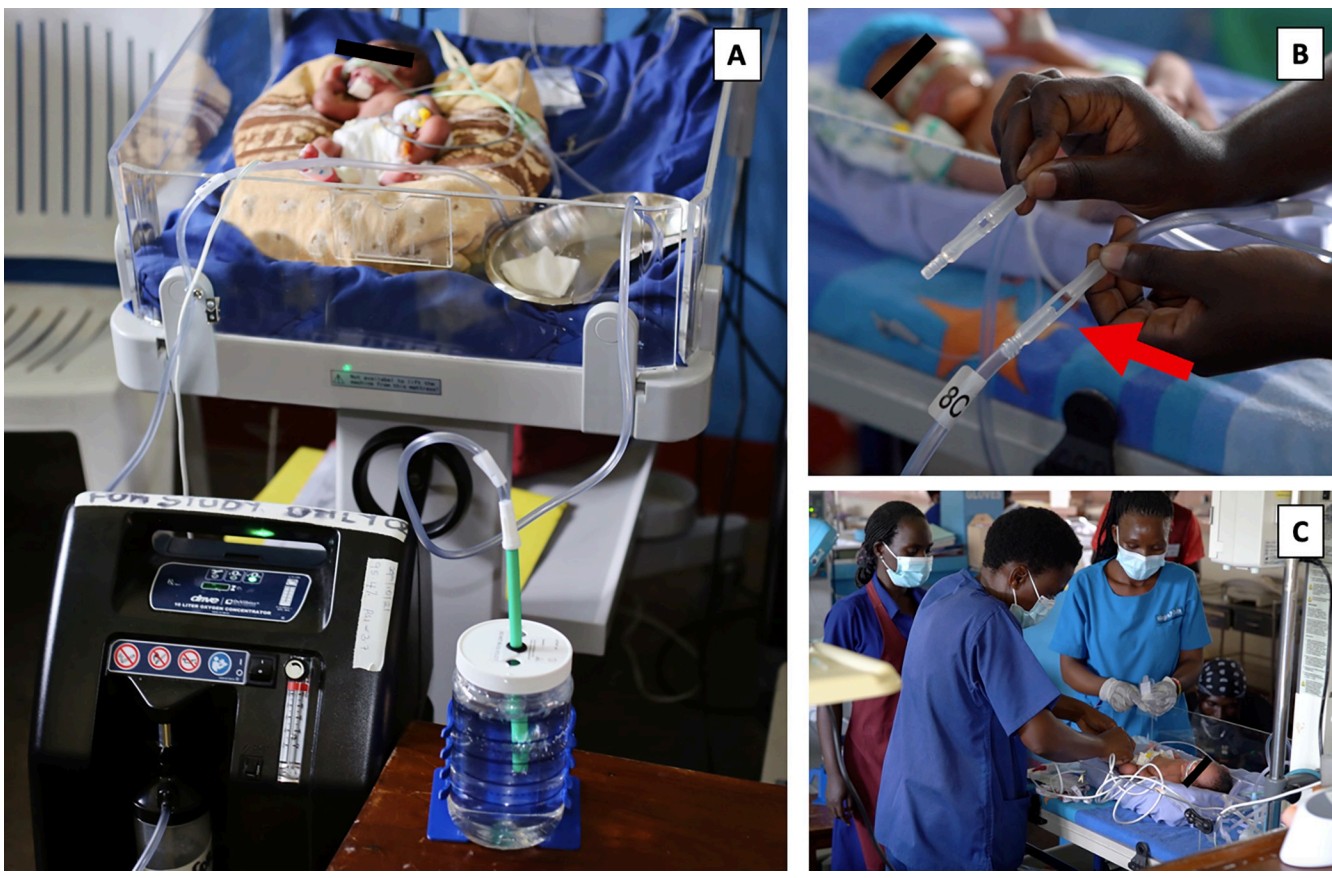

**Fig 3. Neonate on bCPAP circuit.** (A) supplied with flow from an oxygen concentrator. (B) Blender (red arrow) is in place in-line. (C) Healthcare workers caring for a patient and assessing feasibility of the design. Photos used with healthcare worker and parental consent.

## Nursing time and device set-up

The median census in the unit was 46 patients (range 33–60). Median nurses on each shift was 5 (range 3–12). Median patients on bCPAP were 3 (range 0–9). Median patient to nurse ratio in the unit at times of bCPAP initiation was 9.2 (IQR 8.2–10.3). Device set-up took a median of 15 minutes (IQR 12–20). Initial set-up was done correctly 64% of the time (and otherwise corrected with feedback from the research team). Most common difficulties in set-up included finding the correct cannula size (8/11), securing cannula to the face (3/11) and achieving proper saturations with blenders (1/11). Median time to change the blender was 15 seconds (IQR 12–27).

## Healthcare worker experience

A likert-type survey was filled out by 9 nurses/midwives after they set-up and initiated a study participant on the bCPAP device. Five of these respondents went on to set up the device on another study patient and completed a repeat survey. All first time and subsequent user scales recorded (14/14) agreed that the device is easy to set up and the blenders were easy to learn to use. (data in S2 Table). Five of 9 of respondents setting up the device for the first time reported that HCWs at other sites might have challenges using the device, however on subsequent set-ups respondents unanimously believe that other HCWs will not have challenges. 5 of 5 HCW who used the device more than once believed the bCPAP device improves the quality of care

**Table 2. Interview themes among thirteen nurses/midwives (Healthcare workers, HCW) and two Doctors (D).**

| Interview themes (frequency mentioned) | Quotes |
|---|---|
| **Usability** | |
| The device is easy to use (14/15) | *"It can be operated by one person without the need to call for an assistant, and when you want to change the blenders, it does not take a lot of time and it does not compromise the supply of oxygen to the baby."* (HCW6) |
| There is a need for a way to blend oxygen and air for bCPAP therapy. (12/15) | *"But if you do not have (a way to blend oxygen and air) and the baby needs CPAP. . .you are giving excess oxygen to the baby, which is dangerous."* (HCW3) |
| The device is easy to set up (11/15) | *"It is simple to connect, and you can even connect it alone without help. . ."* (HCW11) |
| The tubing connections were complex (8/15) | *"Maybe the only difficult part was mastering where each tubing should go. Like when there is an emergency; where you have divided minds, you might end up setting it up wrongly. . .So, connecting the tubings to the right direction and even the blenders is key. . ."* (HCW4) |
| **Acceptability** | |
| HCW would like to continue use after the study. (15/15) | *"Obviously yes, because it is easier to use. . .the whole device is simple!"* (HCW4) |
| The bCPAP device and blenders is packaged well (14/15) | *"I liked the packaging because it is easily accessed with all the parts together and it is good to use in emergencies."* (HCW9) |
| There is a need for a blender that provides less than 37% (14/14) | *"I was limited on changing the oxygen amount that the baby was getting to 37% and 60%. That was difficult because. . .the baby was getting more oxygen and yet I couldn't reduce the oxygen anymore."* (HCW6) |
| Important components to include with the device:<br>• Quality adhesive to secure prongs to patient's face (13/15)<br> • Nasal foam pads (11/15)<br> • Nasal sizing guide (11/15) | *"It would be impossible [to use the device without supplied tape] . . .obviously the nasal prongs will be coming out all the time and the baby will not be getting the CPAP continuously as it is supposed to be."* (HCW3)<br>*"[without the foam pads] there would be leakage of air and damage to the nasal septum. So, it is really important to have them."* (HCW6)<br>*"Yes, [the nasal sizing guide] should be kept in the device because it helps us get the right nasal prongs for the baby after measuring the nostrils. . .a person can bring three nasal prongs of different sizes to try on the baby to find one that fits. This is time consuming!"* (HCW13) |
| The device is a good option because it provides less than 100% oxygen (8/15) | *"It has helped us to provide blended air and oxygen to all babies who need it, most especially the extremely premature babies."* (HCW10)<br>*"In a setting where you are using just oxygen, this is a massive advantage, and I would recommend it."* (D01) |

they could offer their patients. Overall satisfaction with the device and blenders was 8.5/10 (IQR 6.5–9.5).

Fifteen healthcare workers were interviewed after using the bCPAP device. Of these, thirteen were nurses or midwives with a median of 5 years (IQR 4–7) years working in the newborn care unit, and two were doctors with 2 and 6 years of experience in the newborn care unit. The doctors did not have hands-on experience setting up the bCPAP device but rather spoke about the shared experience obtained through observation of it being used and clinical oversight of the infants enrolled in the research study. All staff interviewed had used bCPAP and blended oxygen previously.

Qualitative themes derived from interviews are described in Table 2. Overall, health care workers demonstrated high acceptability of the device and considered device usability to be reasonably good. Suggested device modifications include additional choices for the fixed blenders and improvements in patient interfaces and securement devices.

## Discussion

We demonstrate use of a novel very low-cost bCPAP device including oxygen blenders is feasible in a Ugandan level two newborn unit and acceptable to healthcare workers with experience with bCPAP and oxygen blending. The nursing time required for setting up the device (15 minutes) and changing blenders (15 seconds) were low and comparable to those reported with a commercialized device [37]. This allows nurses to maintain their attention to their patients and makes the device practical for a busy unit such as where this study was performed. Nurses felt the device was easy to set-up and use, improved care of their patients and they appreciated the opportunity to minimize oxygen toxicity.

The 14 patients treated with the device in accordance with unit bCPAP guidelines were preterm infants (gestation 25–35 weeks) with primarily respiratory distress syndrome. They were initiated on bCPAP near birth and treated for a median of 3 days. Despite delivery of blended oxygen, each patient continued to have periods of hyperoxia (saturations >95%). This hyperoxia also occurred when these patients were on nasal cannula oxygen before bCPAP initiation. The mortality rate (14%) during the study is difficult to interpret given the small sample size, but was below the unit's historical rates on bCPAP per unit records (34%). Overall these outcomes are consistent with other settings lacking advanced respiratory therapies such as surfactant and mechanical ventilation.

Strengths of this study include constant observation of care required for study participants by the research team as well as use of multiple instruments to triangulate and confirm the utility of the device to the healthcare team. Generalizability of this unit's experience with the device is augmented by a broad set of inclusion criteria which did not restrict use of bCPAP from extremely preterm infants nor those with birth asphyxia [17, 38]. As well, the median patient-to-nurse ratio of 9.7 was consistent with other African units administering CPAP suggesting similar nurse workload [9].

Limitations to this study's generalizability include device implementation and assessment of acceptability and feasibility in a single center with experience using a similar bCPAP device, blending oxygen, and saturation targeting. Implementation in units without this experience would require more training, bedside reinforcement and assessment of healthcare worker impression [28]. Within the interview data, interpretation could suffer from a lack of confirmability since no member checks were conducted with the study population and no field notes recording non-verbal observations were taken during the interviews.

Device limitations include the lack of heated humidification and use of a higher resistance, non-sealing nasal interface (the RAM cannula) than other higher priced devices on the market [24, 39–42]. Importantly, however, both features may allow easier bedside device and patient management in settings with high patient-to-nurse ratios. The use of a nasal pad with the RAM cannula interface decreases pressure leak and may also minimize risk of septal damage compared with other interfaces [43, 44]. The occurrence of hyperoxia during the study and interview responses confirm this device would be improved with an additional blender below the 37% range to provide the minimal oxygen required for each patient. Finally, this was not a study of device effectiveness, and no evaluation was made of the effectiveness of achieving the specified percent oxygen or PEEP.

The reality for much of the world's newborns is poor access to bCPAP and, when available, risk of toxicity and developmental impairment from receipt of 100% oxygen. As recognized by the 2030 target established by the Every Newborn Action Plan, availability of bCPAP devices and oxygen blenders are crucial components in advancing quality care required for preterm infants [4, 23]. The provision of respiratory support, however, cannot occur in a vacuum- and settings must ensure concurrent robust provision of thermoregulation, nutritional support

and infection management as well [45]. bCPAP implementation requires training, clinical mentorship and post-implementation support as well as adequate and consistent staffing [8, 14, 28, 46, 47]. For sites ready for bCPAP implementation where higher priced devices are out of reach, further study on the practicality and implementation of very low cost devices including the PATH bCPAP device may be ideal. Scale-up and access to more affordable bCPAP devices with the ability to blend oxygen are needed to achieve World Health Organization targets and improve facility-based care.

## Supporting information

**S1 Checklist. CONSORT checklist for feasibility trial.**
(PDF)

**S1 File. Kiwoko bCPAP guidelines of care.**
(PDF)

**S2 File. Device instructions for use.**
(PDF)

**S3 File. Interview guide.**
(PDF)

**S4 File. Interviewer positionality and content analysis framework.**
(PDF)

**S5 File. Trial protocol.**
(PDF)

**S6 File. ClinicalTrial.gov protocol.**
(PDF)

**S1 Table. Study dataset.**
(XLS)

**S2 Table. Healthcare worker usability responses.**
(PNG)

## Acknowledgments

Kiwoko research assistants and data entry teams, team members Mike Eisenstein, Jill Sherman-Konkle and Manjari Quintanar Solares, MD, MPH
    Qualitative work- Anna Cunningham, BSN, MPH
    Medical monitoring team- Clare Nakubulwa MBChB (MUST), Flaviah Namiiro MBChB, MMED (Paed) and Sheillah Bagayana SBM

## Author Contributions

**Conceptualization:** Anna B. Hedstrom, James Nyonyintono, Eugene A. Saxon, Madeline Vaughan, Alec Wollen, Peter Waiswa, Maneesh Batra.

**Data curation:** Anna B. Hedstrom, Beatrice Niyonshaba.

**Formal analysis:** Anna B. Hedstrom, Heidi Nakamura, Noelle Simpson, Paul Mubiri.

**Funding acquisition:** Anna B. Hedstrom, James Nyonyintono, Eugene A. Saxon, Madeline Vaughan, Alec Wollen, Maneesh Batra.

**Investigation:** Anna B. Hedstrom, James Nyonyintono, Alec Wollen, Patricia S. Coffey.

**Methodology:** Anna B. Hedstrom, James Nyonyintono, Eugene A. Saxon, Heidi Nakamura, Beatrice Niyonshaba, Josephine Nakakande, Alec Wollen, Paul Mubiri, Peter Waiswa, Patricia S. Coffey, Maneesh Batra.

**Project administration:** Anna B. Hedstrom, James Nyonyintono, Heidi Nakamura, Hilda Namakula, Beatrice Niyonshaba, Madeline Vaughan, Paul Mubiri, Patricia S. Coffey.

**Resources:** Madeline Vaughan, Patricia S. Coffey.

**Software:** Paul Mubiri.

**Supervision:** Anna B. Hedstrom, James Nyonyintono, Heidi Nakamura, Hilda Namakula, Beatrice Niyonshaba, Josephine Nakakande, Madeline Vaughan, Patricia S. Coffey, Maneesh Batra.

**Validation:** Hilda Namakula, Alec Wollen, Paul Mubiri.

**Writing – original draft:** Anna B. Hedstrom, Heidi Nakamura, Noelle Simpson, Paul Mubiri, Patricia S. Coffey.

**Writing – review & editing:** James Nyonyintono, Eugene A. Saxon, Madeline Vaughan, Peter Waiswa, Maneesh Batra.

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
