## [Decision Letter · Decision Letter 0]

19 Dec 2022

PGPH-D-22-01392

Feasibility and Usability of a Very Low-Cost Bubble Continuous Positive Airway Pressure Device including Oxygen Blenders in a Ugandan Level Two Newborn Unit

Dear Dr. Hedstrom,

Thank you for submitting your manuscript to PLOS Global Public Health. After careful consideration, we feel that it has merit but does not fully meet PLOS Global Public Health’s publication criteria as it currently stands. Therefore, we invite you to submit a revised version of the manuscript that addresses the points raised during the review process.

EDITOR:

We have received two detailed reviews from expert reviewers, both raising similar comments about the pre-clinical testing of this device (the entire bCPAP circuit, not just the blender) and requesting additional detail. Please provide as much detail as you can on this testing in supplemental material (especially if it has not been previously published in peer-reviewed literature) and make clear in the manuscript itself what aspects have been tested, and what has not. 

We appreciate that this article is focused on the feasibility and usability of this set-up - not on technical efficacy. But typically technical efficacy will have been demonstrated previously and this is not clearly communicated at present. If the device has not been fully tested or does not meet industry standards, please be transparent about what stage it is at with testing and validation. You may also wish to reflect on this in your discussion and conclusions, considering the assurances you would want to be able to use and recommend this set-up to others in the future (i.e. it has demonstrated feasibility and usability, but what else is important).

The reviewers also highlight some areas you can been more measured in your conclusions, clearer about comparison populations and reporting results, and provide sensible suggestions for improving readability and clarity to readers. I encourage you to take their suggestions on board as you revise.

We look forward to receiving your revised manuscript.

Kind regards,

Hamish R Graham

Academic Editor

Journal Requirements:

1. Please remove any photos of children (Figures 2) from your submission.

Additional Editor Comments (if provided):

Reviewers' comments:

Reviewer's Responses to Questions

**Comments to the Author**

1. Does this manuscript meet PLOS Global Public Health’s publication criteria? Is the manuscript technically sound, and do the data support the conclusions? The manuscript must describe methodologically and ethically rigorous research with conclusions that are appropriately drawn based on the data presented.

Reviewer #1: Partly

Reviewer #2: Yes

2. Has the statistical analysis been performed appropriately and rigorously?

Reviewer #1: Yes

Reviewer #2: N/A

3. Have the authors made all data underlying the findings in their manuscript fully available (please refer to the Data Availability Statement at the start of the manuscript PDF file)?

Reviewer #1: Yes

Reviewer #2: Yes

4. Is the manuscript presented in an intelligible fashion and written in standard English?

Reviewer #1: Yes

Reviewer #2: Yes

5. Review Comments to the Author

Reviewer #1: Please see attached file. 

Reviewer #2: Before being applied to patients this research should have had a technical/benchtop peer reviewed engineering paper. The background refers to references 25 and 26 but these are not peer reviewed publications (one a meeting abstract and the other a media piece). Do we even know that this bCPAP System provides flows, oxygen concentrations, pressures, resistance, and imposed work of breathing in ranges that are technically acceptable/reliable? What about humidification?

With only 14 enrolled patients, a survey of 9 providers, with the confounder of a compressor that was used for priority patients, after use/implementation over 4 months in a referral facility newborn unit for 3 Districts (with a median census of 46) makes statements on mortality ("mortality rate below historical") and scale (".....the PATH bCPAP device may be ideal") an inappropriate stretch. Instead, this is a very early peek into a novel device which prompts a number of intriguing questions that inspire further study. It is far too early to suggest this is ready for use on patients given the information/data to-date.

6. PLOS authors have the option to publish the peer review history of their article (what does this mean?). If published, this will include your full peer review and any attached files.

**Do you want your identity to be public for this peer review?** For information about this choice, including consent withdrawal, please see our Privacy Policy.

Reviewer #1: No

Reviewer #2: No

---

## [Editor Report · Decision Letter 1]

8 Feb 2023

Feasibility and Usability of a Very Low-Cost Bubble Continuous Positive Airway Pressure Device including Oxygen Blenders in a Ugandan Level Two Newborn Unit

PGPH-D-22-01392R1

Dear Dr. Hedstrom,

We are pleased to inform you that your manuscript 'Feasibility and Usability of a Very Low-Cost Bubble Continuous Positive Airway Pressure Device including Oxygen Blenders in a Ugandan Level Two Newborn Unit' has been provisionally accepted for publication in PLOS Global Public Health. Thank you for your patience as we obtained Editorial and Reviewer feedback to make this decision.

Best regards,

Hamish R Graham

Academic Editor